# Preparation of Doxorubicin-Loaded Carboxymethyl-β-Cyclodextrin/Chitosan Nanoparticles with Antioxidant, Antitumor Activities and pH-Sensitive Release

**DOI:** 10.3390/md20050278

**Published:** 2022-04-21

**Authors:** Yingqi Mi, Jingjing Zhang, Wenqiang Tan, Qin Miao, Qing Li, Zhanyong Guo

**Affiliations:** 1Key Laboratory of Coastal Biology and Bioresource Utilization, Yantai Institute of Coastal Zone Research, Chinese Academy of Sciences, Yantai 264003, China; yqmi@yic.ac.cn (Y.M.); jingjingzhang@yic.ac.cn (J.Z.); wqtan@yic.ac.cn (W.T.); qmiao@yic.ac.cn (Q.M.); qli@yic.ac.cn (Q.L.); 2University of Chinese Academy of Sciences, Beijing 100049, China

**Keywords:** chitosan nanoparticles, cyclodextrin, antioxidant activity, antitumor activity, drug release

## Abstract

In this study, chitosan nanoparticles (HF-CD NPs) were synthesized by an ionic gelation method using negatively charged carboxymethyl-β-cyclodextrin and positively charged 2-hydroxypropyltrimethyl ammonium chloride chitosan bearing folic acid. The surface morphology of HF-CD NPs was spherical or oval, and they possessed relatively small particle size (192 ± 8 nm) and positive zeta potential (+20 ± 2 mV). Meanwhile, doxorubicin (Dox) was selected as model drug to investigate the prepared nanoparticles’ potential to serve as a drug delivery carrier. The drug loading efficiency of drug-loaded nanoparticles (HF-Dox-CD NPs) was 31.25%. In vitro release profiles showed that Dox release of nanoparticles represented a pH-sensitive sustained and controlled release characteristic. At the same time, the antioxidant activity of nanoparticles was measured, and chitosan nanoparticles possessed good antioxidant activity and could inhibit the lipid peroxidation inside the cell and avoid material infection. Notably, CCK-8 assay testified that the nanoparticles were safe drug carriers and significantly enhanced the antitumor activity of Dox. The nanoparticles possessed good antioxidant activity, pH-sensitive sustained controlled release, enhanced antitumor activity, and could be expected to serve as a drug carrier in future with broad application prospects.

## 1. Introduction

The ever-increasing morbidity and mortality induced by different cancers have developed a severe worldwide problem which is constantly impacting the social medical security system of human society [1,2,3]. Currently, chemotherapy with cytotoxic drugs remains one of the major therapy methods for clinical cancer therapy [4]. However, clinical data showed that there are two challenges of chemotherapy in clinical application. One of the challenges is that chemotherapeutic drugs are highly toxic and can damage healthy tissues while exerting anti-tumor effects [5]. Multiple drug resistance (MDR) is also a large challenge to chemotherapy by reducing the sensitivity of tumor cells to chemotherapeutic drugs [2,6]. Taking doxorubicin (Dox) for example, as a kind of common broad-spectrum antitumor drug, it is an anthracycline antibiotic obtained by separation from *streptomyces peucetius var. caesius* [7]. The DNA damage caused by blocking replication of DNA is the mechanism of achieving the anti-cancer effect of Dox [3]. However, as described above, there are some problems with the clinical use of Dox. On one hand, data on the clinical application of Dox illustrates that the “burst effect” of Dox release can increase the toxic and side effects and liver nephridium burden [7,8]. On the other hand, the development of MDR leads to the poor bioavailability and pharmacokinetics [9,10]. Therefore, it is extremely important to develop new drug delivery systems aiming at reducing toxic/side effects and MDR in chemotherapy.

In recent years, nanoencapsulation of chemotherapy drugs by natural biopolymers has drawn considerable interest from researchers, and it is considered as a breakthrough in reducing toxic/side effects and reversing MDR of chemotherapy [4,11]. Meanwhile, it is reported that nanoencapsulation of chemotherapy drugs possesses many advantages such as the avoidance of drug’s premature degradation, enhancement of absorption into target tissue, increase in action time and bioavailability, and improvement in intracellular penetration [4,12]. Generally, it is recommended that nanoscale drug carriers should possess certain properties such as good biocompatibility, nontoxicity, no immunity, and acceptable biodegradation time, which can protect chemotherapy drugs at the target site and prolong the drug efficacy [13,14]. Hence, chitosan is considered as a potential nanomaterial because it precisely satisfied the requirements of nanoscale drug carriers.

Chitosan, a kind of cationic polysaccharide, composed of β-(1,4)-linked *N*-acetyl-d-glucosamine, is obtained by partial or complete deacetylation of chitin which is extracted from shrimp and crab shells, insects, and fungi [15,16,17]. Chitosan bears many hydroxyl and amino groups, which can be easily chemically modified for further application [18]. Thereinto, 2-hydroxypropyltrimethyl ammonium chloride chitosan (HACC), a chitosan derivative prepared by introducing the quaternary ammonium group with good hydration ability has drawn more interest from researchers [16]. It is reported that HACC possesses many biological properties, including nontoxicity, antimicrobial, anti-inflammatory, and antioxidant activities, which can be considered as promising nanoscale drug carriers in drug delivery systems [19]. Meanwhile, carboxymethyl-β-cyclodextrin (CMCD), a cyclic oligosaccharide with a cone-shaped structure composed of seven glucose units that possesses a hydrophobic cavity and hydrophilic shell at the outer surface, has been attracting the attention of pharmacists [11,20]. It is reported that CMCD could form inclusion complexes by encapsulating a variety of hydrophobic guest molecules such as drugs and proteins in the hydrophobic cavity through non-covalent interactions [4]. Numerous studies have demonstrated that the incorporation of CMCD in nanoparticles for drug delivery of various bioactive compounds such as naringenin, warfarin, curcumin, calcium folinate, methotrexate, and glutathione could enhance solubility and stability, control drug release, improve bioavailability, increase drug loading capacity, and reduce toxicity [5,11,14]. Considering that CMCD belongs to a kind of polyanion compound, nanoparticles can be synthesized by an ionic gelation method using negatively charged CMCD and positively charged HACC. In addition, folic acid (FA) is a kind of water-soluble vitamin that possesses high bioactivity and low immunogenicity. Many studies have shown that FA could serve as the targeting ligand due to the overexpression of folate receptors on the surface of many cancer cells [3]. Therefore, we hypothesized that CMCD/HACC bearing FA nanoparticles prepared by ionic gelation without the introduction of other chemical crosslinking agents might serve as a potential Dox carrier with the advantages of reduced toxicity, significantly increased drug loading efficacy, extended drug release time, and enhanced efficacy.

The purpose of this study is to prepare drug-loading nanoparticles via the ionic gelation technique using negatively charged CMCD and positively charged HACC bearing FA in a mild condition and to research the potential of drug-loading nanoparticles. The physicochemical property of nanoparticles was thoroughly characterized by Fourier transform infrared spectroscopy (FTIR), ^1^H Nuclear magnetic resonance spectroscopy (^1^H NMR), zeta potential measurement, dynamic light scattering (DLS), and scanning electron microscopy (SEM). Meanwhile, the encapsulation efficiency (EE), drug loading capacity (DLE), and in vitro release analysis under three different pH values (6.8, 7.4, and 8.0) of drug-loading nanoparticles were investigated. Subsequently, the antioxidant efficiency against DPPH and superoxide anion radicals of nanoparticles was investigated. The antitumor activity against human gastric carcinoma cell line (BGC-823), human breast carcinoma cell line (MCF-7), human hepatocellular carcinoma cell line (HEPG-2), and human lung cancer cell line (A549) was performed by CCK-8 assay in vitro. Finally, the cytotoxicity was also measured by CCK-8 assay against mouse fibroblast cell line (L929) to probe the biocompatibility of nanoparticles.

## 2. Results and Discussion

### 2.1. Characterization of Chitosan Derivatives and Chitosan Nanoparticles

Chitosan nanoparticles, both pure and Dox loaded, were synthesized by an ionic gelation method using negatively charged CMCD and positively charged HACC bearing FA (HF) (Figure 1). The physicochemical property of nanoparticles was thoroughly characterized by FTIR, ^1^H NMR, nanometer particle size measuring instrument, and SEM.

#### 2.1.1. FTIR Spectra Analysis

The structures of chitosan, chitosan derivatives, CMCD, and prepared nanoparticles were confirmed by FTIR as shown in Figure 1. For unmodified chitosan, the major characteristic absorptions appear at 3416 cm^−1^ (the OH and NH vibrations), 2876 cm^−1^ (the C-H vibration), 1655 cm^−^^1^ (the C=O stretching vibration of the residual amide bond), and 1080 cm^−^^1^ (the C-O stretching) [18]. For HACC, the absorption band at 1479 cm^−1^ (stretching vibration of N^+^(CH_3_)_3_) appears in the spectrum, which indicates the successful introduction of a quaternary ammonium group [17]. As for HF, after the introduction of folic acid anion, new characteristic absorptions appear at 1507 cm^−1^ (the C=C stretching for the benzene ring in folate molecules). Compared to CD, the new characteristic peak at 1730 cm^−1^ (COOH stretching vibration) in the spectrum of CMCD can demonstrate the successful carboxymethylation reaction [4,11]. After ionic gelation, although the characteristic absorption appears at 1477 cm^−1^ of N^+^(CH_3_)_3_, a new peak at 1030 cm^−1^ assigned to the C-O stretching vibration of the CD molecule appears. As for HF-CD NPs, although the characteristic absorption appears at 1477 cm^−1^ of N^+^(CH_3_)_3_, a new peak at 1513 cm^−1^ assigned to the C=C stretching vibration of the benzene ring in the folate molecule is also observed [3]. In comparison with HF-CD NPs spectrum, the addition of Dox increased the intensity of the NH and CH stretching bands at 3410 cm^−1^ in the spectrum of HF-Dox-CD NPs. Meanwhile, the new characteristic absorption appeared at 1027 cm^−1^ (the C-O stretching of Dox), representing the presence of Dox in the chitosan nanoparticle, the same characteristic peaks also found in H-Dox-CD NPs [9,19]. Hence, the structures of chitosan, chitosan derivatives, CMCD, and prepared nanoparticles were preliminarily confirmed by FTIR.

#### 2.1.2. ^1^H NMR Spectra Analysis

The structures of chitosan derivatives and CMCD were further investigated by ^1^H NMR spectroscopy. As shown in the spectrum of HACC (Figure 2a), the characteristic signal at 3.12 ppm (d) was assigned to the protons of N^+^(CH_3_)_3_. At the same time, other characteristic peaks located at 2.57 ppm (c), 2.82 ppm (a), 4.20 ppm (b), are easy to observe in the spectrum of HACC, which can further prove the successful introduction of the quaternary ammonium group [21]. After the introduction of folic acid anions by ion exchange, new peaks appeared at 6.34 ppm (i, j) and 7.40 ppm (k, l) in the HF spectrum, attributed to the protons of the benzene ring. Additionally, the characteristic signal at 3.10 ppm representing the protons of N^+^(CH_3_)_3_ also can be observed from HF [19]. In the spectrum of CMCD, the obvious peak located at 4.11 ppm is assigned to the protons of the carboxymethyl groups (CH_2_COOH), and the singlet peak that appeared at 5.00 ppm is attributed to the anomeric protons [22]. These ^1^H NMR analyses illustrated the successful modification of chitosan derivatives and CMCD. Furthermore, the DS of chitosan derivatives and CMCD are calculated by ^1^H NMR. Additionally, the DS of HACC, HF, and CMCD are 67.00%, 48.00%, and 57.50%, respectively. Considering that HACC and HF need a certain number of cations for further ionic gelation, appropriate DS is important. Taking HF as an example, the DS of 48% means that the ratio between folic acid and cations in HF molecules is close to 1:1, that is to say, half of the cations can carry out the further reaction with CMCD. Meanwhile, after carboxymethylation, the carboxylate anions of CMCD can be crosslinked with HACC or HF to prepare nanoparticles. Therefore, the appropriate DS of chitosan derivatives and CMCD is critical for the further reaction.

#### 2.1.3. Hydrodynamic Diameter (nm), Polydispersity Index (%), and Zeta Potential (mV)

As the vital factors, the hydrodynamic diameter, polydispersity index (PDI), and zeta potential determined the stability, drug loading, and delivery efficiency of nanoparticles [14,16,17]. Figure 3 shows the hydrodynamic diameter and PDI of chitosan nanoparticles. From the figure, there are some rules that can be summarized as follows. Firstly, all prepared chitosan nanoparticles have relatively small particle size and uniform distribution. Specifically, the hydrodynamic diameters of H-CD NPs, HF-CD NPs, H-Dox-CD NPs, HF-Dox-CD NPs are 156 ± 7, 192 ± 8, 220 ± 15, and 222 ± 12 nm, respectively. Secondly, after loading the drug, the hydrodynamic diameter of chitosan nanoparticles (H-Dox-CD NPs and HF-Dox-CD NPs) increases significantly, but still does not exceed 500 nm. Moreover, the particle size distribution intensity of chitosan nanoparticles is shown in Table 1. It can be seen that the particle size of the nanoparticles without drug loading is mostly distributed between 100 and 200 nm, and after drug loading, the particle size is mostly distributed between 200 and 300 nm. In short, all the nanoparticles are less than 500 nm in size. Numerous studies have shown that nanoparticles with particle size of 50–500 nm will be taken by endocytosis rather than phagocytosis [2,23]. Chitosan nanoparticles prepared in the paper are less than 500 nm, illustrating that the nanoparticles can be endocytosed by cells. Thirdly, the PDI indexes of prepared chitosan nanoparticles are less than 0.24. It is worth noting that the lower PDI represents the more uniform particle size distribution [16].

As a necessary standard to evaluate the stability of colloidal dispersions, zeta potential refers to the electrostatic repulsion existing in colloidal dispersions. Generally, the stability of colloids is related to zeta potential [14,17]. Figure 4 shows the zeta potential of chitosan nanoparticles. From the figure, chitosan nanoparticles all possess a positive zeta potential ranging from 19 to 21 mV. Moreover, there are numerous papers that have shown that nanoparticles possessing positive surface charge are easily absorbed by cells due to the negative charge on the cell membrane [16,24]. Therefore, prepared nanoparticles possessing positive zeta potential are more easily absorbed by cells.

#### 2.1.4. Morphology Analysis and Nanoparticle Stability Analysis

The morphology of H-CD NPs, HF-CD NPs, H-Dox-CD NPs, and HF-Dox-CD NPs were conducted by SEM as shown in Figure 5a. From the figure, the obtained nanoparticles are seen to be spherical or almost spherical, which is consistent with the hydrodynamic diameter measured by a nanometer particle size measuring instrument. For HF-Dox-CD NPs obtained by ionic gelation method, CMCD and HF assembled into the spherical or almost spherical shape with the Dox inlaid into the spherical shell as a nucleus [11]. Additionally, this can also explain the larger size of nanoparticles after drug loading. Meanwhile, after Dox loading, prepared nanoparticles are easy to agglomerate in water, which leads to the “coating phenomenon” for the relatively weak dispersion of these agglomerates [25]. Therefore, in addition to spherical or almost spherical nanoparticles, some of the drug loading nanoparticles are aggregated like fibrous particles. Moreover, the nanoparticles stability is evaluated and shown in Figure 5b. From the figure, the particle size of nanoparticles does not change significantly as time goes on, illustrating the good stability of nanoparticles.

### 2.2. Entrapped Efficiency (EE) and Drug Loading Efficiency (DLE) Analysis

The EE and DLE of nanoparticles with Dox loading including H-Dox-CD NPs and HF-Dox-CD NPs were measured under the optimal conditions. H-Dox-CD NPs and HF-Dox-CD NPs possess EE of 73.25% and 75.75%, respectively. Additionally, DLE of H-Dox-CD NPs and HF-Dox-CD NPs are 30.08% and 31.25%, respectively. It was worth noting that the EE and DLE of DOX in the nanoparticles with Dox loading were improved and raised compared with a previous report [19]. It can be attributed to the fact that CMCD could effectively embed hydrophobic drugs into the tapered cavity to increase the EE and DLE [4]. CMCD possessed pH sensitivity and unique hydrophobic cavity and hydrophilic shell structure, which can effectively embed hydrophobic drugs into the hollow truncated conical cavity. A similar observation has been reported by Oluwatobi et al. [26]. In other words, chitosan nanoparticles prepared with CMCD as polyanionic crosslinking agent have significantly improved EE and DLE, which is consistent with our expectation.

### 2.3. In Vitro Release of Chitosan Nanoparticles

In vitro release studies of free Dox, H-Dox-CD NPs, and HF-Dox-CD NPs were carried out using PBS with different pH values (pH = 8.0, 7.4, 6.8) as the release medium. Some rules can be summarized from Figure 6. Firstly, as previously reported, the release of free Dox under different pH conditions presents a “burst effect”, which may enhance cytotoxicity, shorten action time, and increase the burden of liver and kidney [3,27]. Specifically, free Dox is released very rapidly in the first 4 h, with a cumulative release of 67.21% (pH = 6.8). However, free Dox is released very slowly over the following 44 h. Secondly, it is obvious that the release of Dox from CMCD/chitosan nanoparticles can be divided into three phases based on the rate of release. Taking Figure 6a, for example, the first phase refers to the initial phase where burst release of Dox can be discovered during the first 4 h. The cumulative release percentages of HF-Dox-CD NPs and H-Dox-CD NPs in buffer solution with pH 6.8 are up to 22.02% and 25.32%, respectively. The mechanism of release in the first phase is attributed to the release of Dox molecules adsorbed on or near the surface of nanoparticles [26,28]. The release of Dox at this stage is mainly influenced by diffusion driving force. When nanoparticles bind to water, hydrogen bonds form on the surface of nanoparticles to accelerate drug release [28]. The second phase that occurred between 4 and 12 h is controlled release where the release rate decreases slightly. As time goes on, the cumulative release percentages in this phase of HF-Dox-CD NPs in buffer solution with pH 6.8 are 22.02%, 27.81%, and 35.45% at 4, 8, and 12 h, respectively. The relatively slow release rate of Dox in this phase might be due to the controlled diffusion through the gel-like layer formed on the nanoparticle surface. It was reported that the drug diffusion from chitosan nanoparticles behaved as a hydrogel [26]. When chitosan nanoparticles were exposed to water, the nanoparticles might swell and release the drugs by diffusion. The third phase that occurred between 12 and 48 h is sustained or controlled release where the release rate decreased obviously compared to the second phase. In this stage, the release of Dox is by diffusion of the medium into the nanoparticulate system, which led the disintegration of the polymer matrix [29]. Thirdly, when the pH values of the dissolved medium are 6.8, 7.4, and 8.0, the cumulative release rates of DOX from HF-Dox-CD NPs are 53.75%, 48.46%, and 45.06% at 48 h, respectively. That is, the lower the pH of the releasing medium, the higher the cumulative release rate is. In vitro release studies demonstrated that the Dox release from chitosan nanoparticles was highly pH-sensitive, which might be attributed to two reasons. On one hand, chitosan nanoparticles swelling tended to be high at the low pH condition, which could accelerate the drug release [25]. On the other hand, under the acidic condition, the protons might penetrate the inside and attack the inner secondary bonds of nanoparticles, which changed the conformation of nanoparticles [26]. It is worth noting that the pH of tumor tissue might reduce from 7.4 (normal tissue) to under 6.8 (pathological tissue) due to the acids secreted by cancer cells [19]. Hence, the CMCD/chitosan nanoparticles possessed a good controlled release effect and pH-sensitive drug release, which could serve as a controlled drug delivery system for Dox.

### 2.4. Antioxidant Activity Analysis

The antioxidant activity of chitosan, chitosan derivatives, CMCD, Dox, as well as chitosan nanoparticles was evaluated using the superoxide-radical and DPPH radical scavenging assay in vitro. Figure 7a shows the superoxide-radical scavenging effect of chitosan, chitosan derivatives, CMCD, Dox, and chitosan nanoparticles. From the figure, some rules can be summarized as follows: Firstly, the antioxidant activity is concentration-dependent for all tested samples. For example, the superoxide-radical scavenging indexes of HF-Dox-CD NPs are 78.24%, 70.43%, 61.92%, 55.54%, and 45.52% when the sample concentrations are 1.6, 0.8, 0.4, 0.2, and 0.1 mg/mL, respectively. Secondly, chitosan and Dox show almost no antioxidant activity. However, after the introduction of folic acid anions by ion exchange, HF shows a significantly enhanced superoxide-radical scavenging index of 68.07% at 1.6 mg/mL. It has been reported that folic acid played an important role in antioxidant activity by destroying active oxidizing radicals with aid of thermodynamically favorable and reversible oxidation, which was consistent with this paper [3,19]. Thirdly, HF-Dox-CD NPs possesses the best superoxide-radical scavenging index of 78.24% at 1.6 mg/mL. The similar rules can be found in Figure 7b. Briefly, the obvious decreasing trend in DPPH radical scavenging activity is found for HF-Dox-CD NPs, HF-CD NPs, H-Dox-CD NPs, H-CD NPs, and Dox with the decreased antioxidant activities of 69.04%, 64.81%, 51.40%, 50.32%, and 30.48% at 1.6 mg/mL. In short, compared to Dox, the chitosan nanoparticles prepared in this paper possessed much stronger antioxidant activity. It is worth noting that good biological activity like antioxidant activity is the key factor to avoid material infection and immune reaction [20]. Hence, HF-Dox-CD NPs nanoparticles possessed good antioxidant activity that could inhibit the lipid peroxidation inside the cell and avoid material infection, which might serve as a potential drug carrier.

### 2.5. Cytotoxicity Analysis

The antitumor activity of chitosan, chitosan derivatives, CMCD, chitosan nanoparticles, and free Dox against different cancer cells (BGC-823, MCF-7, HEPG-2, and A549) was performed by CCK-8 assay in vitro. It is worth noting that the lower cell viability of test samples means higher antitumor activity. From Figure 8a, the CMCD/chitosan nanoparticles with Dox loading including H-Dox-CD NPs and HF-Dox-CD NPs have antitumor activity against BGC-823 cancer cells. Briefly, compared to free Dox, the CMCD/chitosan nanoparticles with Dox loading possess much stronger antitumor activity. For example, for HF-Dox-CD NPs the cell viability at 320 μg/mL is 9.33%, which is far lower than the cell viability of free Dox (15.57%). Meanwhile, the cell viability of free Dox and CMCD/chitosan nanoparticles with Dox loading are negative concentration dependent. In other words, with the increase in concentration of the tested sample, the sample possessed lower cell viability. Taking HF-Dox-CD NPs as an example, when the series of concentrations are 0, 10, 20, 40, 80, 160, and 320 μg/mL, the cell viabilities are 104.34%, 63.15%, 28.20%, 12.56%, 10.31%, 9.70%, and 9.33%, respectively. The similar rules can be summarized in Figure 8b. Compared to free Dox, H-Dox-CD NPs and HF-Dox-CD NPs possess much stronger antitumor activity against MCF-7 cancer cells. It is worth noting that HF-Dox-CD NPs has the minimum cell viability of 24.91% at the maximum concentration, meaning the strongest antitumor activity. The cell viability against HEPG-2 and A549 cells were also measured, and results show that the CMCD/chitosan nanoparticles with Dox loading possess much stronger antitumor activity than free Dox. Therefore, combining Figure 8a–d, some rules can be found that both CMCD/chitosan nanoparticles with Dox loading possessed significantly improved antitumor activity than free Dox against different cancer cells (BGC-823, MCF-7, HEPG-2, and A549). Furthermore, the half-maximal inhibitory concentration (IC_50_) values of drug-loading nanoparticles and Dox against different cells can also illustrate that the antitumor activity of the drug-loading nanoparticles including H-Dox-CD NPs and HF-Dox-CD NPs is higher than the free Dox. For example, the IC_50_ against BGC-823 cells shows that the values of H-Dox-CD NPs and HF-Dox-CD NPs are 12.0 and 12.2 μg/mL, respectively, while the value of free Dox is 41.2 μg/mL (Table 2). The enhanced antitumor activity of chitosan nanoparticles might be due to the small particle size and endocytosis, which could promote the permeability of drug into the cancer cells [26]. Additionally, the synergistic efficiency of drug carrier and Dox was confirmed by the significantly enhanced antitumor activity of nanoparticles, which was consistent with a previous report [3]. Meanwhile, the antitumor activity decreased following the order of HF-Dox-CD NPs > H-Dox-CD NPs > Dox, which might be attributed to the bioactivity of folic acid. Numerous studies have demonstrated that folic acid was highly effective at targeting tumor cells as a ligand due to the over-expression of the folic acid receptor on the human cancer cell surface [3,19]. Finally, the cell viability of free Dox and drug loading nanoparticles was negatively concentration dependent against different cancer cells (*p* < 0.05). On balance, the self-activity of drug carrier and synergistic efficiency with Dox increased the antitumor activity.

In vitro cytotoxicity assay is an essential procedure to evaluate the intracellular drug delivery and biosafety of nanoparticles. In this study, the cytotoxicity of tested samples against L929 cells was measured by CCK-8 and the results are shown in Figure 9. For chitosan, the results show negligible cytotoxicity at all tested concentrations against L929 cells, which is also confirmed by the cell growth diagram (Figure 10). Specifically, the normal cell morphology is spindle or oval, and the less normal cells mean strong cytotoxicity. Meanwhile, the cell viability of blank nanoparticles (H-CD NPs and HF-CD NPs) against L929 cells is about 75–100%, which suggests the good biocompatibility of blank nanoparticles. However, the cell viability of free Dox is as low as 33.07% at maximum concentration (1000 μg/mL), indicating the high cytotoxicity. What is noteworthy is that the cytotoxicity is significantly reduced when the drug was encapsulated in the nanoparticles. Moreover, the cytotoxicity of drug loaded nanoparticles and free Dox showed a dose-dependent relationship, which was consist with previous research [2,6]. Therefore, these results indicated that prepared chitosan nanoparticles with significantly enhanced antitumor activity and reduced cytotoxicity might possess broad application prospects in the field of future clinical chemotherapy.

## 3. Materials and Methods

### 3.1. Materials

Chitosan with the molecular weight of 200 k Da and the deacetylation degree of 85% was obtained from Qingdao Baicheng Biochemical Corp. (Qingdao, China). β-cyclodextrin (CD) with 1.1k Da MW and 95% purity was purchased from Bidepharm (Shanghai, China). Doxorubicin hydrochloride and FA were purchased from Aladdin Industrial Corp. (Shanghai, China). The chemical reagents including absolute ethyl alcohol acetone, and isopropyl alcohol were obtained from Sinopharm Chemical Reagent Co., Ltd. (Shanghai, China). The materials, including hydrochloric acid (HCl), sodium hydroxide (NaOH), chloroacetic acid, 2, 3-epoxypropyl trimethyl ammonium chloride (ETA), and 2,2-diphenyl-1-picrylhydrazyl (DPPH) radical were obtained from Sigma-Aldrich Chemical Corp. (Shanghai, China). Human tumor cell lines (BGC-823, MCF-7, HEPG-2, and A549) and fibroblasts (L929) were provided by Luye Pharma Group (Yantai, China). All the above chemical solvents and reagents used in this study were analytical grades.

### 3.2. Preparation of Nanoparticles

#### 3.2.1. Preparation of CMCD

CMCD was synthesized from CD according to previous reports with some relatively minor modifications [22,30]. In short, the CD powder (1.42 g, 10 mmol) was introduced in a round-bottomed flask containing 16 mL of isopropanol and stirred at 300 rpm for 1 h in order to mix evenly. Then, 5 mL of NaOH solution (40%, *w*/*v*) was dropwise added to the CD solution. The activation was carried out in a metal bath at 50 °C with stirring at 300 rpm for 1 h. Subsequently, 27 mL of chloroacetic acid solution (10%, *w*/*v*) was dropwise added to the above solution and the solution was stirred vigorously at 65 °C for 5 h. The CMCD was gained by precipitation with acetone, washing with alcohol, and lyophilization to constant weight.

#### 3.2.2. Preparation of HACC and HF

HACC was prepared by modifying the previous preparation method [14]. Briefly, chitosan (1.61 g, 10 mmol) dispersed in 80 mL isopropanol with the mechanical stirring lasted for 4 h at 75 °C. Then, 30 mL of ETA solution (60%, *w*/*v*) was dropwise added to the prior reaction system and reacted at 75 °C with the stirring speed of 300 rpm for 12 h. After adjusting the pH of the reaction mixture to 7 by appropriate volume of HCl aqueous solution (10%, *w*/*v*), the HACC was obtained by dialysis with distilled water and lyophilization to constant weight.

Folic acid (4.41 g, 10 mmol) was mixed with NaOH (0.4 g, 10 mmol), and 80 mL deionized water was added to it with stirring in order to obtain a solution of sodium folate. HACC (3.12 g, 10 mmol) was dripped into the prior sodium folate solution with the stirring speed of 300 rpm at 25 °C. After 12 h of reaction, the unreacted anions including excess folate, chlorine, and sodium ions were removed by dialysis (molecular weight cutoff: 500 Da) against deionized water for 48 h at 25 °C. The residual solution in the dialysis tube was dried in a vacuum to obtain HF.

#### 3.2.3. Preparation of Nanoparticles Based on HACC and HF

HACC loading Dox nanoparticles (H-Dox-CD NPs) were synthesized by an ionic gelation method using negatively charged CMCD and positively charged HACC. The applicable reaction condition was explored based on particle size, zeta potential, drug loading effect, and encapsulation efficiency through experiments. Briefly, 0.3 g HACC was dissolved in 150 mL of deionized water at 25 °C while stirring overnight. After sterilizing using a 0.22 μm filter, 30 mL of doxorubicin hydrochloride solution (2 mg/mL) was dropwise added to HACC solution. Then, 1 h later, CMCD aqueous solution (37 mL, 2 mg/mL) was added dropwise to the prior solution while stirring at 800 rpm at 25 °C for 30 min. H-Dox-CD NPs were separated by centrifugation at 12,000× *g* rpm for 20 min. Finally, the purified and dry chitosan nanoparticles were obtained by vacuum freeze drying. Meanwhile, pure nanoparticles (H-CD NPs) without doxorubicin hydrochloride were prepared following the above method, but Dox solution was replaced by the same volume of deionized water.

Similarly, for HF loading Dox nanoparticles (HF-Dox-CD NPs), the appropriate preparation condition was explored by experiments based on particle size, electric potential, drug loading effect, and other factors. In short, 100 mL of HF (2 mg/mL) was dissolved in deionized water at 25 °C while stirring overnight. After sterilizing using a 0.22 μm filter, 30 mL of doxorubicin hydrochloride solution (2 mg/mL) was dropwise added to HF solution. Then, 1 h later, CMCD aqueous solution (40 mL, 2 mg/mL) was added dropwise to the prior solution while stirring at 800 rpm at 25 °C for 30 min. HF-Dox-CD NPs were separated by centrifugation at 12,000× *g* rpm for 20 min. Finally, the purified and dry chitosan nanoparticles were obtained by vacuum freeze drying. Meanwhile, pure nanoparticles (HF-CD NPs) without Dox loading were prepared following the above method, but doxorubicin hydrochloride solution was replaced by the same volume of deionized water. The nanoparticles were stored sealed in a dry environment for subsequent testing.

### 3.3. Characterization

#### 3.3.1. Fourier Transform Infrared Spectroscopy (FTIR)

In order to determine the structures of chitosan, chitosan derivatives, CMCD, and chitosan nanoparticles, the FTIR spectra were investigated by KBr pellet method on a FTIR spectrometer (Nicolet iS50, Thermo, Waltham, MA, USA) over the frequency ranging from 500 to 4000 cm^−1^ with a resolution of 4 cm^−1^.

#### 3.3.2. ^1^H Nuclear Magnetic Resonance Spectroscopy (^1^H NMR)

The chitosan, chitosan derivatives, CD, and CMCD (10 mg/mL) were dissolved in D_2_O and measured by a Bruker AVIII-500 Spectrometer (500 MHz, Switzerland, provided by Bruker Tech. and Serv. Co., Ltd., Beijing, China) at 25 °C. Subsequently, the degrees of substitution (DS) were measured in accordance with the integral ratio of different hydrogen protons [31]. For example, the DS of CMCD was calculated by the following formula:(1)$$\mathrm{DS}\left(\%\right)=\frac{\frac{I_{H_{2,5,7,}CMCD}-2I_{H_{1,}CMCD}}{2}}{I_{H_{1,}CMCD}}\times 100$$
where $I_{H_{2,5,7,}CMCD}$ means the total integral values of the H of CH_2_COOH; $I_{H_{1,}CMCD}$ means the integral value of the hydrogen atom bonded to Carbon 1 of cyclodextrin backbone; 2 represents the number of protons in H_7_ of CMCD.

#### 3.3.3. Hydrodynamic Diameter (nm), Polydispersity Index (%), and Zeta Potential (mV)

The hydrodynamic diameter, polydispersity index (PDI), and zeta potential of the NPs were measured using a Nanometer particle size measuring instrument (Litesizer 500, Anton Paar Instruments, Graz, Austria). The data were presented as the mean ± standard deviation (SD) according to three independent measurements.

#### 3.3.4. Morphology and Nanoparticle Stability

The morphology of prepared nanoparticles was recorded by scanning electron microscopy (SEM, S-4800, Hitachi, Tokyo, Japan). In short, the prepared nanoparticles were placed on a sample-holder coated with gold for observation by SEM.

Meanwhile, in order to explore the stability of nanoparticles, particle size of nanoparticles was measured three times at specific time intervals at room temperature using the Nanometer particle size measuring instrument. The data were presented as the mean ± standard deviation (SD).

### 3.4. Entrapped Efficiency (EE) and Drug Loading Efficiency (DLE) of Nanoparticles

In order to explore the inclusion ability of nanoparticles, the EE and DLE were measured using the UV–vis spectroscopic method [3,16]. Briefly, prepared nanoparticles solution was centrifuged at 12,000× *g* rpm for 20 min. Then, the free Dox concentration in supernatant was calculated through the UV-vis spectrophotometer (T6, Pgeneral, Beijing, China) at 480 nm. Finally, the total mass of Dox in supernatant was calculated according to the free Dox concentration. The experiment was repeated three times, and EE and DLE were obtained as the following equations:(2)$$\mathrm{EE}\;(\%)=\frac{m_{\;total\;Dox}-m_{\;free\;Dox}}{m_{\;total\;Dox}}\times 100$$
(3)$$\mathrm{DLE}\;(\%)=\frac{m_{\;total\;Dox}-m_{\;free\;Dox}}{m_{\;total\;NP}}\times 100$$

In the formula, $m_{\;total\;Dox}$ represents the total quantity of Dox added, $m_{\;free\;Dox}$ represents the quantity of free Dox in supernatant, and $m_{\;total\;NP}$ represents the quantity of nanoparticles.

### 3.5. In Vitro Release Study

To investigate the Dox release behavior, the release percentage was measured by dissolution tester (RC1207DP, Tianjin, China) and UV-vis spectrophotometer (T6, Pgeneral, Beijing, China) using PBS with different pH values (pH = 8.0, 7.4, 6.8) as the release medium to simulate the internal environment [26]. Briefly, 10 mL of Dox-loaded nanoparticles solution (5 mg/mL) was transferred to a dialysis tube, and the dialysis tube was suspended, respectively, in 800 mL of phosphate buffer with different pH values (pH = 8.0, 7.4, 6.8) with moderate stirring (100 rpm) at 37 °C. At predetermined intervals, 2 mL of release media was extracted, and replaced with equal volume of fresh PBS immediately. The released Dox was calculated in accordance with the standard curve of the absorbance (A) and concentration (C) through the UV–vis spectrophotometer (T6, Pgeneral, Beijing, China) at 480 nm. The release percentage was measured by following equation:(4)$$\mathrm{Release}\;\mathrm{percentage}\;(\%)=\sum_{t=0}^{t}\frac{m_{t}}{m_{0}}\times 100$$

In the formula, $m_{\;t}$ means the quantity of released Dox at predetermined time, and $m_{\;0}$ means the initial quality of Dox.

### 3.6. Antioxidant Assays

#### 3.6.1. Superoxide-Radical Scavenging Activity Assay

The superoxide-radical scavenging ability was measured following a previous method [19]. Firstly, different samples were dissolved in deionized water to ensure the original concentration of 10 mg/mL. Then, a series of volumes of the sample solutions (0.96, 0.48, 0.24, 0.12, and 0.06 mL) were introduced to the test tube, adding phenazine methosulfate (PMS, 30 μM), nicotinamide adenine dinucleotide reduced (NADH, 338 μM), and nitro blue tetrazolium (NBT, 72 μM) in Tris-HCl buffer (16 mM, pH 8.0). Subsequently, the sample solution was shaken evenly and incubated at 25 °C in the dark for 5 min. Finally, the absorbance was measured quickly at 560 nm and each experiment was repeated three times. The superoxide-radical scavenging effect was calculated by the following equation:(5)$$\mathrm{Scavenging}\;\mathrm{effect}\;\left(\%\right)=\left[1-\frac{A_{sample\;560\;\mathrm{nm}}-A_{control\;560\;\mathrm{nm}}}{A_{blank\;560\;\mathrm{nm}}}\right]\times 100$$
where $A_{sample\;560\;\mathrm{nm}}$ means the absorbance of samples, $A_{control\;560\;\mathrm{nm}}$ represents the absorbance of the control group (distilled water was instead of NADH), and $A_{blank\;560\;\mathrm{nm}}$. represents the absorbance of the blank group (the sample was replaced by distilled water).

#### 3.6.2. DPPH Radical Scavenging Ability Assay

The DPPH-radical scavenging activity assay was measured following the previous report with a slight modification [20]. Firstly, different dried samples (40 mg) were dissolved in 40 mL of deionized water in order to get the original concentration of 10 mg/mL, respectively. Then, a series of volumes of the sample solutions (0.48, 0.24, 0.12, 0.06, and 0.03 mL) were introduced to the test tube, with DPPH ethanol solution added (2 mL, 180.0 μmol/L). The test samples with final sample solution concentrations of 1.6, 0.8, 0.4, 0.2, and 0.1 mg/mL were incubated in the dark at 25 °C for 20 min. At the same time, in the blank group, the sample was replaced with 1 mL of deionized water, and in the control group, the DPPH ethanol solution was instead replaced with 2 mL of ethyl alcohol. Finally, the absorbance of the sample solution at 517 nm was measured with a microplate reader (DNM-9602G, Thermo Multiskan Ascent, MA, USA). The experiment was repeated three times for each sample, and the DPPH radical scavenging ability was calculated as follows:(6)$$\mathrm{Scavenging}\;\mathrm{effect}\;\left(\%\right)=\left[1-\frac{A_{sample\;517\;\mathrm{nm}}-A_{control\;517\;\mathrm{nm}}}{A_{blank\;517\;\mathrm{nm}}}\right]\times 100$$
where $A_{sample\;517\;\mathrm{nm}}$ means the absorbance of samples, $A_{control\;517\;\mathrm{nm}}$ means the absorbance of the control, and $A_{blank\;517\;\mathrm{nm}}$ means the absorbance of the blank.

### 3.7. Cytotoxicity Assay

The antitumor activity against different cancer cells (BGC-823, MCF-7, HEPG-2, and A549) was performed by CCK-8 assay in vitro [25]. Meanwhile, the cytotoxicity was also measured by CCK-8 assay on L929 cells to probe the biocompatibility of nanoparticles. Briefly, the different cells in the logarithmic phase were seeded in 96-well flat-bottom culture plates and incubated in 100 µL of RPMI medium at the specific incubator (5% CO_2_, 37 °C) for 24 h. The cells were respectively treated for 24 h with fresh medium containing chitosan, chitosan derivatives, CMCD, nanoparticles, and free DOX. After incubation, every well was injected with 10 µL of CCK-8 solution and incubated for another 24 h. The absorbance at 450 nm was recorded with the microplate reader (DNM-9602G, Thermo Multiskan Ascent, USA) to calculate the cell viability:(7)$$\mathrm{Cell}\;\mathrm{viability}\;\left(\%\right)=\frac{A{\;}_{sample}-A_{\;blank}}{A_{negative}-A_{blank}}\times 100$$
where $A{\;}_{sample}$ is the absorbance of samples (containing cells, CCK-8 and sample solution), $A_{\;blank}$ represents the absorbance of blank (containing RPMI medium and CCK-8 solution), and $A_{negative}$ means the absorbance of negative (containing cells and CCK-8 solution).

### 3.8. Statistical Analysis

All experiments were repeated in triplicate (n = 3), and the data were shown as mean value ± standard deviation (SD) for each group. All experimental data were evaluated using one-way analysis of variance (ANOVA) followed by Duncan multiple comparison. The level of *p* < 0.05 was considered statistically significant.

## 4. Conclusions

In this study, novel CMCD/chitosan nanoparticles with Dox loaded were synthesized by an ionic gelation method using negatively charged CMCD and positively charged HF. The physicochemical property of nanoparticles was thoroughly characterized by FTIR, ^1^H NMR, nanometer particle size measuring instrument, and SEM. Results showed that the surface morphology of HF-Dox-CD NPs was spherical or oval, and they possessed relatively small particle size (222 ± 12 nm) and positive zeta potential (+19 ± 3 mV). The nanoparticle stability was determined and showed that the particle size of nanoparticles did not change significantly as time passed, illustrating the good stability of nanoparticles. It is worth noting that chitosan nanoparticles prepared with CMCD as the polyanionic crosslinking agent have significantly improved EE and DLE. In vitro release profiles indicated that the CMCD/chitosan nanoparticles possessed controlled and pH-sensitive drug release, which could serve as a controlled drug delivery system for Dox. At the same time, the antioxidant activity of nanoparticles was measured, and chitosan nanoparticles possessed good antioxidant activity and could inhibit the lipid peroxidation inside the cell and avoid material infection. Notably, CCK-8 assay testified that the nanoparticles were safe drug carriers and significantly enhanced the antitumor activity of Dox. The nanoparticles possessed good antioxidant activity, pH-sensitive sustained controlled release, and enhanced antitumor activity; therefore, they could be expected to serve as a drug carrier in future with broad application prospects.

## Data Availability

All data are contained in the manuscript.

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
