# Peer review of "Preparation of Doxorubicin-Loaded Carboxymethyl-β-Cyclodextrin/Chitosan Nanoparticles with Antioxidant, Antitumor Activities and pH-Sensitive Release"

_marinedrugs, 2022, doi:10.3390/md20050278_

Round 1

Reviewer 1 Report

The paper proposes the use of nanoparticles obtained from a chitosan derivative and a cyclodextrin derivative as a drug-delivery system for an antitumor drug: doxorubicin. The nanoparticles also contain folic acid as a targeting ligand and to bring antioxidant activity.

A previous paper of the same research group [ref. 17] show that HACC, folic acid and carboxymethyl chitosan form polyelectrolyte complexes nanoparticles that can be loaded with doxorubicin and have antioxidant and antitumor activity. What is the novelty of the present paper given by the replacement of carboxymethyl chitosan with carboxymethyl cyclodextrin (CMCD)? What is the difference between the hypothesis form ref 17 (Carbohydrate Polymers 2021, 273, 118623.) and the hypothesis from the present paper.

               If CMCD form an inclusion complex with doxorubicin (Dox), as presented in the Scheme 1, this should be demonstrated. In the literature it was shown that the cavity size of β-cyclodextrin is too small to include the lipophilic aglycone part of doxorubicin [O. Bekers, J.H. Beijnen, M. Otagiri et all., Inclusion complex of doxorubicin and daunorubicin with cyclodextrins, Journal of Pharmaceutical & Biomedical Analysis, 1990, 8, 671-674], but some derivatives of β-cyclodextrin were shown to form complexes with Dox [https://doi.org/10.1021/jp2091363].

               In the Scheme 1, the loading with Dox was presented after the formation of the HF -CMCD nanoparticles, but in the description of the methods, Dox was added before CMCD.

From the NMR spectra of CMCD, the signal of H-7 is overlapped with the signal of the protons H2-6, so H7 cannot be very precisely integrated. The DS can be calculated from the integrals of H1 and of H2-H7 protons (3.3-4.3 ppm): DS(%) = (IH2-H7 -5IH1)/2 *100. The DS can be also determine by the potentiometric or conductometric titration of the –COOH groups. If the DS for CMCD is 48%, it means that around 3 glucose subunits (0.48*7=3.3) of the CD are substituted with carboxymethyl? The structure f CMCD from Scheme 1 can be modified accordingly.

               The reaction of CS with 3-chloro-2-hydroxypropyltrimethyl ammonium chloride at basic pH cannot take place at –OH groups of CS? The number of cationic groups (-NH2 and tertiary amine groups) per gram CS can be determine by electrochemical titration.

What is the ratio between folic acid and glucosamine units of CS in HF? For the further interaction with CMCD it is important to know how many cationic groups remain free.

 Chapter 3.2.3: The amount of Dox, or the concentration of the Dox solution should be mentioned. What is the molar ratio between the cationic groups of HACC (or HF) and the carboxylic groups of CMCD? Probably the ratio of cationic groups: anionic groups was higher than 1 to assure the positive zeta potential of the nanoparticles.

Figure 9: The cell viability decrease with the decrease of the polymer concentration (or nanoparticles concentration)? In reference 17 the results are opposite, as it should be.

Pag 2, line 84: CMCD/HACC was not grafted (chemically) with folic acid.

Pag 11, line 341: Doxorubicin is in the hydrochloride form?

Pag 11, line 353: The references shoud be introduced as numbers.

Author Response

Please find attached the cover letter for you.

Reviewer 2 Report

In this study, HF-CD NPs were synthesized by an ionic gelation method using negatively charged carboxymethyl-β-cyclodextrin and positively charged 2-hydroxypropyltrimethyl ammonium chloride chitosan grafting folic acid. The physicochemical property of nanoparticles was characterized by different techniques. The surface morphology of NPs was spherical or oval, and they possessed small particle size and positive Z potential. The authors found that NPs possessed good antioxidant activity, pH-sensitive sustained controlled release of Dox, and enhanced antitumor activity. Although the experimental part seems to me correctly performed, the paper needs, in my opinion, a major revision before to be accepted for publication.

- Throughout the work: size, zeta potential and PDI should be expressed with the appropriate significant figures. Examples:

                 222±12 nm and NOT  222.02±12.21 nm

                 +19±3 mV and NOT +19.44±2.78 mV

                 0.18±0.02 and NOT 0.18±0.023

- Lines 99-101: The different cell lines must be specified.

- Section 2.1.1: The results of H-CD NPs and H-dox-CD NPs should be discussed.

- Lines 145-146: the results of DS of chitosan derivatives and CMCD should be indicated.

- Section 2.2: The discussion of the EE and DLE results is very poor, in fact, they simply show them. The authors should improve the discussion of these results.

- Section 2.3: The authors do not mention the results of the release at pH 7.4. why did they do it then? They should discuss these results or else they should remove it from the paper.

- Figures 7 and 9 b need to be improved, they are of poor quality.

- Figure 9 a: In this figure the authors have made a mistake in the legends, it is impossible that cell viability is less at higher concentrations of NPs or dox than at lower concentrations of them.c

- Lines 385 and 395: The international acronym for revolutions per minute is rpm. Authors use a very strange one such as r/min and sometimes mix the two, rpm/min, as in these lines and it is redundant. I advise authors to always use rpm.

- Sections 3.3.2, 3.4, 3.5, 3.6.2 and 3.7: Equations should be numbered.

Author Response

Dear Reviewer:

Please find the cover letter for you in the attachment.

Reviewer 3 Report

The authors present the use of chitosan derived nanomaterials for the delivery of dox in various types of cancer cells. Here are some comments

  1. MW and DDA of Chitosan-this is crucial to understand the size of the particles and their stability
  2. Size distribution from DLS-include the % number values and curves.
  3. Zeta potential values don not indicate high stability. The sizes are also much larger and undesirable for drug delivery applications-especially in low vasculature tumors.
  4. Re-check PDI values-don’t seem to match with the curves. The particle size numbers are also ambiguous. The curves show they range between 100-1000 yet the average diameters are reported as 150-200nm
  5. TEM images and histograms required for better understanding.
  6. In vitro release: seems like there is no difference in the release of dox at pH 6.8 and 7.4. This is undesirable and provides no advantage. Further, this indicates release of Dox at the physiological pH which would increase the toxicity-Dox is known for its cardiotoxicity already.
  7. Provide comparison of IC50 of free vs encapsulated Dox on the cell lines
  8. Cellular uptake studies of Dox loaded NPs needed-a few time points to see the uptake and efflux of the particles.
  9. Following references should be added:

https://pubmed.ncbi.nlm.nih.gov/11516503/

https://pubmed.ncbi.nlm.nih.gov/33151075/

Author Response

Dear Reviewer,

Please find the cover letter for you in the attachment.

Round 2

Reviewer 1 Report

The paper was improved. 

One small aspect must be corrected: line 370, CTA remained in the Materials section, instead of ETA.

Author Response

The paper was improved.

One small aspect must be corrected: line 370, CTA remained in the Materials section, instead of ETA.

Answer: Thank you for your kind suggestions and according to your recommendation, we have replaced 3-chloro-2-hydroxypropyl trimethyl ammonium chloride (CTA) with 2, 3-epoxypropyl trimethyl ammonium chloride (ETA), and marked the revised manuscript in red (Line 369). Very sorry for this mistake. Thank you for your kind suggestions and we hope meet with approval.

Reviewer 2 Report

Minor points

Line 308: authors says "Furthermore, the IC50 values of..." and they shoud say "Furthermore, the half-maximal inhibitory concentration, IC50, values of..."

Author Response

Minor points

Line 308: authors says "Furthermore, the IC50 values of..." and they shoud say "Furthermore, the half-maximal inhibitory concentration, IC50, values of..."

Answer: Thank you for your kind suggestions and according to your recommendation, we have corrected the sentence and marked it in red (Lines 316-318). Very sorry for this mistake. Thank you for your kind suggestions and we hope meet with approval.

Reviewer 3 Report

Authors have tried addressing most concerns within their capability. Some experimental data still remains questionable (large particle size, PDI, drug release at pH 7.4, cellular uptake etc. )

Author Response

Authors have tried addressing most concerns within their capability. Some experimental data still remains questionable (large particle size, PDI, drug release at pH 7.4, cellular uptake etc. )

Answer: Thank you for your kind suggestions. The main purpose of the research was to develop a kind of high drug-loaded nanoparticle, so the carboxymethyl-β-cyclodextrin was utilized in the preparation of nanoparticles. Accordingly, the evaluation of encapsulation and drug loading efficiency, drug release behavior, and in vitro antioxidant and antitumor activities were involved in this article. In the future, more studies will be conducted to evaluate the practicability of this nanoparticle in the nano drug delivery system, such as cell uptake experiments and animal experiments. The PDI was measured three times by nanometer particle size measuring instrument (Litesizer 500, Anton Paar Instruments, Graz, Austria). In addition, the diameters reported as 150-200 nm in the manuscript represented the average particle size, and the specific particle size distribution was also given in Table 1. Thank you for your kind suggestions and we hope meet with approval.